# Divergent Effects of Daidzein and Its Metabolites on Estrogen-Induced Survival of Breast Cancer Cells

**DOI:** 10.3390/cancers12010167

**Published:** 2020-01-09

**Authors:** Emiliano Montalesi, Manuela Cipolletti, Patrizio Cracco, Marco Fiocchetti, Maria Marino

**Affiliations:** Department of Science, University Roma Tre, Viale Guglielmo Marconi 446, I-00146 Roma, Italy; emiliano.montalesi@uniroma3.it (E.M.); manuela.cipolletti@uniroma3.it (M.C.); patrizio.cracco@uniroma3.it (P.C.); marco.fiocchetti@uniroma3.it (M.F.)

**Keywords:** estrogen, estrogen receptor alpha, polyphenols, daidzein, daidzein metabolites, paclitaxel, apoptosis, breast cancer cells

## Abstract

Although soy consumption is associated with breast cancer prevention, the low bioavailability and the extensive metabolism of soy-active components limit their clinical application. Here, the impact of daidzein (D) and its metabolites on estrogen-dependent anti-apoptotic pathway has been evaluated in breast cancer cells. In estrogen receptor α-positive breast cancer cells treated with D and its metabolites, single or in mixture, ERα activation and Neuroglobin (NGB) levels, an anti-apoptotic estrogen/ERα-inducible protein, were evaluated. Moreover, the apoptotic cascade activation, as well as the cell number after stimulation was assessed in the absence/presence of paclitaxel to determine the compound effects on cell susceptibility to a chemotherapeutic agent. Among the metabolites, only D-4′-sulfate maintains the anti-estrogenic effect of D, reducing the NGB levels and rendering breast cancer cells more prone to the paclitaxel treatment, whereas other metabolites showed estrogen mimetic effects, or even estrogen independent effects. Intriguingly, the co-stimulation of D and gut metabolites strongly reduced D effects. The results highlight the important and complex influence of metabolic transformation on isoflavones physiological effects and demonstrate the need to take biotransformation into account when assessing the potential health benefits of consumption of soy isoflavones in cancer.

## 1. Introduction

Plant-derived polyphenols are naturally occurring nonsteroidal compounds that play important roles in ecological functions such as pollinator attraction or protection from herbivores and UV irradiation [1]. Due to their molecular structure and size, some of these molecules, including lignans, flavonoids, and stilbenes, have a chemical structure that resembles that of human estrogens, in particular to 17-β-estradiol (E2) [2]. Among other, isoflavones, a class of flavonoids ranked among the most estrogenic compounds, bind to estrogen receptor subtypes (i.e., ERα and ERβ) [3,4] exerting estrogenic and/or antiestrogenic effects [1]. For isoflavones, the key to their bioactivity in human and animals seems to rely on their (anti)estrogenic activity. Indeed, due their antiestrogenic activities, isoflavone enriched diets are associated with a lower incidence of a variety of estrogen-related cancers, including breast, endometrial, and ovarian cancers [5,6]. 

The main dietary source of isoflavones in humans are soybean and soybean products, which contain mainly daidzein (7,4′-dihydroxyisoflavone, D) and genistein (7, 4′-dihydroxy-6-methoxyisoflavone), whose potential efficacy against breast cancer is well documented [7,8,9,10,11,12]. Although these data are promising for the use of these compounds as anticancer therapeutic agents, the therapeutic application of isoflavones is still limited, mainly due to their scarce bioavailability in human beings. In particular, D is almost completely metabolized by the gut microbiota and liver, resulting in water-soluble metabolites (e.g., equol, d-sulfates, o-desmethylangolensin) [13]. Thus, not only D has a low concentration and persistence in the bloodstream, even when consumed in high quantities, but also its metabolites strongly overcome the concentrations of the precursor probably affecting D biological activities [14,15,16,17,18]. Nowadays, the possibility that the metabolites may mimic the anti-carcinogenic effect of D in endocrine-related cancers or may act as synergistic or antagonistic molecules of their precursor is still unknown. Previously, the ability of D, metabolites, to modulate ERβ subtype activities activating a pro-apoptotic cascade in HeLa cancer cells transfected with ERβ expression vector, has been reported [19]. The possibility that a similar scenario could be found also in the modulation of ERα-dependent activities important for breast cancer cell progression is intriguing.

Breast cancer is one of the most common fatal diseases in women. A considerably higher ERα/ERβ ratio is reported in some breast cancer types, when compared to a healthy tissue, namely because of a reduction in the ERβ level [20,21]. The 70% of breast cancers are ERα-positive, where this subtype of receptor mediates E2-induced cancer cell survival and proliferation [22,23,24,25]. In particular, we recently demonstrated that E2 stimulation rapidly enhances the ERα activity (Ser118 phosphorylation and PI3K/AKT pathway activation) in breast cancer cells increasing the intracellular levels of an anti-apoptotic globin, neuroglobin (NGB) [22]. E2-induced NGB upregulation in cancer cells represents an inducible defense mechanism of E2-related human breast cancer rendering them insensitive to several injury including chemotherapy [22,26,27]. Indeed, NGB displays a pivotal role in the E2/ERα-induced anti-apoptotic pathway that abrogates the cell death induced by a chemotherapeutic agent (paclitaxel, Pacl) [22]. Intriguingly, the stilbene Resveratrol decreases NGB levels interfering with E2/ERα-induced NGB up-regulation potentiating Pacl pro-apoptotic effects [4]. In this study, we investigated the potential interference of daidzein on this pathway and evaluated if its metabolites produced mainly from gut microbiota (i.e., equol, Eq, and O-desmethylangolensin, O-DMA) and from both liver and gut enzymes (D-4’-sulfate, D4S, D-7-sulfate, D7S, and D-4’,7-disulfate, DDS) mimic D effect or may act as synergistic or antagonistic molecules. The ERα positive breast cancer cells, MCF-7 and T47D, have been used as the experimental models.

## 2. Results

### 2.1. Effect of D and Its Metabolites on NGB Levels in Breast Cancer Cells

NGB levels were evaluated in MCF-7 cells pre-treated for 24 h with different concentrations of D (Figure 1a) and its metabolites ranging between 0.1 and 10 µM (Figure 1). E2 (10 nM, 24 h) was used as positive control. Eq and O-DMA (Figure 1b) and D7S, D4S, and DDS (Figure 1c) were selected as prototypes of the D metabolites mainly produced by gut microbiota and the liver, respectively. The results clearly indicate that daidzein (1–10 µM) and D4S (0.1–1 µM) reduced the basal level of NGB levels in MCF-7 cells. On the other hand, Eq, O-DMA, D7S, and DDS, like E2, increased the level of NGB (Figure 1a,b). For the successive experiments, D and its mimetic sulphate metabolite (i.e., D4S) were selected. In addition, Eq, one gut metabolite, was selected as negative control. All compounds were used at 1 µM concentration in successive experiments.

The modulation of NGB levels by D, D4S, and Eq (1 µM, 24 h) was also confirmed in T47D cells (Figure 2). Indeed, also in these ERα-positive cells, D and D4S significantly reduced the basal level of NGB, whereas Eq, like E2, increased the globin level (Figure 2).

### 2.2. Mechanisms of D-, D4S-, and Eq Induced Modulation of NGB Levels

The involvement of ERα in the effects of D and its metabolites has been confirmed by pre-treating MCF-7 cells with 100 nM of the ERα inhibitor Endoxifen (Endo) before compound stimulation. As shown in Figure 3a, endoxifen pre-treatment completely impairs E2- and Eq-induced NGB up-regulation as well as D- and D4S-induced NGB down-regulation, strongly corroborating the necessity of an active ERα to modulate NGB levels. In particular, E2 rapidly down-regulates ERα levels maintaining high its phosphorylation status (Figure 3b) while neither D nor its metabolites modify the receptor levels but still increase ERα phosphorylation, although at lower level than E2 (Figure 3b). As expected, endoxifen pre-treatment completely prevents the ERα activation by all compounds considered (Figure 3b).

ERα activation is the first step of a signal pathway triggered by E2 to enhance NGB levels. The activation of AKT is necessary to rapidly impair NGB degradation and assure NGB gene transcription via the CREBP transcription factor [28]. On the other hand, the ability of flavonoids (i.e., naringenin) to trigger the ERα-dependent activation of p38 has been demonstrated [29]. These evidences prompted us to evaluate if D and its metabolites trigger the activation of these kinases. Figure 4 shows that, as expected, E2 elicits the rapid and persistent activation of AKT enhancing its phosphorylation status in MCF-7 cells after both 1 h and 24 h of stimulation (Figure 4a,b). On the other hand, the hormone rapidly activates p38 phosphorylation (Figure 4c), but 24 h after stimulation, the phosphorylation status of p38 return similar to the control (Figure 4d). Completely different is the effect of D and its metabolites. Indeed, both D and D4S do not activate AKT phosphorylation, but these compounds, in particular D4S, trigger the rapid and persistent activation of p38 phosphorylation (Figure 4c,d). Endoxifen pretreatment prevents the D and D4S effects as well as that of E2, although 1 h after D4S stimulation the ER inhibitor does not completely impede p38 activation (Figure 4c), suggesting that an ERα-independent mechanism is at the root of the very high p38 phosphorylation induced by this sulphate metabolite. Similarly, Eq stimulation of MCF-7 cells rapidly activates AKT phosphorylation by an ERα-independent pathway, not prevented by endoxifen pre-treatment (Figure 4a). However, the Eq-induced AKT activation is transient, and indeed the level of kinase phosphorylation was similar to the control 24 h after Eq stimulation. Like D and D4S, Eq triggers the rapid and persistent ERα-dependent p38 activation (Figure 4c,d). As a whole, these data strongly sustain that daidzein does not share similar action mechanisms with all of its metabolites, at least 1 h after treatment.

### 2.3. Physiological Outcomes of D- and D4S-Induced E2/ERα/NGB Pathway Avoidance

We previously demonstrated that the ability of Resveratrol to impair NGB accumulation rendered cancer cells more prone to the anticancer effect of the chemotherapeutic agent paclitaxel (Pacl) [4]. This evidence prompted us to evaluate if D and D4S also exhibit this ability. As expected, Pacl treatment (100 nM) reduces NGB levels (Figure 5a–c) with the parallel increase of cleaved PARP-1 (i.e., 86 kDa band), a well-known biomarker of late apoptotic events (Figure 5d–f), and reduction of cell number (Figure 5g). Cell pre-treatment with E2 strongly prevents all Pacl effects in MCF-7 cells still enhancing NGB levels (Figure 5a–c), cell number (Figure 5g), and strongly reducing Pacl-induced PARP-1 cleavage (Figure 5d–f). Although neither D nor D4S pre-treatment affected Pacl effects, both these compounds restored Pacl effects in the presence of E2 (Figure 5). The ability of D and D4S to restore Pacl effects on cell number is also confirmed in T47D cell line (Figure 5h).

### 2.4. Physiological Outcomes of D in Mixture with Its Metabolites

Due to its extensive biotransformation in the human body, D in circulation and in tissue is mainly present as a mixture with its metabolites. In order to evaluate if D maintains its anti-estrogenic actions described before, MCF-7 cells were treated with a mixture of compounds containing microbiota-produced metabolites (i.e., Eq and O-DMA, 1 µM each, gut metabolites) or metabolites produced by the gut and in liver enzymes (i.e., D7S, DDS, and D4S, 1 µM each, S metabolites) in presence or absence of D (1 µM). Figure 6a shows that the concentration of mixtures does not exert cytotoxic effects, indeed, the DNA content, and consequently the cell number, remain constant in MCF-7 cells stimulated with mixture or with the single compounds. D maintains the ability to reduce NGB levels only in the co-stimulation with the mixture of sulphate metabolites (S metab), whereas this isoflavone effect is completely impaired by co-stimulation with the gut metabolites (Figure 6b). Notably, the mixture of S metabolites reduces NGB levels with respect to the control, while the mixture of gut metabolites increases NGB levels with respect to the control (Figure 6b). More intriguing is the effect of mixtures on Pacl-induced apoptosis (Figure 6c,d). Like the single compounds (Figure 5), neither sulphate nor gut metabolites induce the PARP cleavage at the concentration tested, however, sulphate metabolites preserve the Pacl-induced PARP-1 cleavage even in the presence of D or of E2 (Figure 6c), while gut metabolites reduce the Pacl effects even in the presence of D (Figure 6d). This latter effect is more evident in the presence of E2 (Figure 6d). As per other experiments, identical results have been obtained in T47D (data not showed), confirming that the mixture effects is not dependent on cellular context. 

## 3. Discussion

Soy-derived isoflavones consumption has been largely recommended to the Western population for their possible vital role in maintaining human health through the regulation of metabolism and body weight, concurrent to the prevention of chronic and degenerative diseases including cancer and neurodegenerative disorders. Still, their short- and long-term effects have not been fully characterized, although some have cautioned that there may be harmful effects of overconsumption, especially in cases where compounds are isolated rather than consumed in a food matrix [30,31]. To further complicate this picture, isoflavones, once consumed as either aglycone or glycosides, enter a complex pathway of biotransformation that renders almost negligible the presence of the original molecule. The relative concentration of different metabolites in both plasma and tissues is determined by the specific contribution of intestinal microbiota and by de-conjugation/conjugation processes within the human body. Nowadays, available data on the estrogenic activity of D metabolites are restricted largely to Eq, whose production depends on the individual ability to host specific intestinal bacteria [32]. Once absorbed, daidzein is efficiently re-conjugated in the gut with either glucuronic acid or sulfate. Conjugation with sulfonic acid takes place also in the liver by hepatic sulfotransferase enzymes. Therefore, the plasma level of isoflavones in people on a soy-rich diet is very low (about 1–5 µM) [5,14], while they are present in the circulation predominantly in their glucuronide and sulfate forms [33].

Nowadays, the ability of D metabolites to maintain the effects of their precursor is largely unknown. The main aim of this study was to determine whether D metabolites produced by sulfotransferase and by microbiota enzymes maintain their anticancer effects, consisting in affecting ERα activities that are important for E2-induced resistance of breast cancer cells to chemotherapeutic injury. For this purpose, we utilized two human ERα positive breast cancer cells in which the pathway E2/ERα/NGB has been previously identified as pivotal for breast cancer cell susceptibility to the chemotherapeutic agent paclitaxel [22]. 

Our data indicate that, unlike E2 that induced NGB overexpression, 1–10 µM D reduced NGB levels under the basal level (i.e., vehicle -treated samples) in both MCF-7 and T47D breast cancer cells. This D effect was mimicked only by the D4S metabolite that reduced NGB levels at lower concentrations than D (i.e., 0.1–1 µM); while Eq and D7S showed an E2 like behavior, increasing NGB levels in a concentration dependent manner. Surprisingly, DDS and O-DMA increased NGB levels only at very low concentrations (i.e., 0.1 µM), and were ineffective at high concentrations (10 µM for DDS and 1 µM for O-DMA, respectively).

The differences between D and its metabolites in modulating NGB levels, as well as the different concentrations necessary to obtain the effect, suggest that D and its metabolites may trigger different signal transduction pathways. D4S and Eq were selected as representatives of the two contrasting effects to determine their mechanisms. Like E2, D, D4S, and Eq trigger ERαS118 phosphorylation, even if only E2 reduces the receptor levels, an important mechanism for E2-induced cancer cell proliferation [34]. Moreover, ERα is necessary for D and its metabolites to regulate NGB levels, confirming that D and its metabolites could bind to and activate ERα, as already reported [3]. However, downstream of ERα activation, D and its metabolites trigger divergent signal pathways. Indeed, differently from E2, D and D4S do not trigger AKT activation, which is pivotal for E2-induced NGB accumulation [28]. Instead, these compounds rapidly (1 h) and persistently (24 h) activate p38 kinase, whose activation is commonly shared among flavonoids (e.g., naringenin, quercetin) [29,35]. Upon receptor binding, naringenin modifies ERα conformation driving the receptor far from the plasma membrane (i.e., receptor de-palmitoylation) and decoupling its association with the active sub-unit of PI3K, but not with p38 kinase [24,29]. Consequently, AKT is not activated, whereas the persistent activation of p38 occurs, driving cancer cells to the activation of the apoptotic cascade that culminates with PARP cleavage [29]. A similar signal transduction pathway seems to be activated by D and D4S that at the concentrations used here (1 µM) do not affect the phosphorylation status of AKT, allowing the persistent p38 activation. On the other hand, Eq rapidly and transiently activates AKT phosphorylation that even if does not impair the persistent p38 activation, as E2 does, is sufficient to accumulate NGB into the cells. Note that the D4S-induced p38 and Eq-induced AKT phosphorylation is ERα-independent, at least partially for D4S, sustaining that these metabolites could bind to other cellular receptors, including the Arylic Receptor [36,37], which, in turn, can interfere with the estrogenic signal. As a whole, the chemical structure of D and its metabolites allows different ERα conformations that, in turn, drive cells to physiological outcomes that differ from that triggered by E2 [24,38,39].

Paclitaxel, a first line therapeutic agent for breast cancer, is a prototype of chemotherapeutic agents, which action mechanism is well known [4]. As reported above, E2/ERα-induced NGB accumulation in cancer cells represents an anti-apoptotic pathway, which abrogates the cell death induced by a chemotherapeutic agent (paclitaxel, Pacl) [4]. The ability of NGB accumulation to act as a shield against Pacl has been further confirmed here. In fact, E2 pretreatment impairs Pacl reduction of NGB as well as its ability to reduce cell number and activate PARP cleavage. Although at the selected concentrations (1 µM) they do not affect cell number or PARP cleavage, D and D4S completely prevent E2 effects, allowing the Pacl-induced activation of a pro-apoptotic cascade, even in the presence of the hormone, sustaining that the anti-estrogenic activity is pivotal for rendering cancer cells more vulnerable to the chemotherapeutic drug. The anti-proliferative action of isoflavones and other plant-derived polyphenols in cancer cells has been widely reported and disputed. Nowadays, it is quite accepted that only high concentrations of isoflavones (≤10 µM), very far from the concentration present in the plasma after a meal rich in polyphenols, could activate a mitochondrial-dependent apoptotic cascade, while low concentrations of these compounds result less active or, even, increase cancer cell proliferation [6,7,8,9]. Our data confirm that low concentrations of D and D4S are unable to trigger apoptosis or enhance Pacl effects, but acting as anti-estrogenic compounds, they can avoid the E2-induced anti-apoptotic effect on this chemotherapeutic drug.

The current study indicates, for the first time, that just D4S but no other metabolites retain the D anti-estrogenic activity. This effect is maintained when cancer cells were co-treated with D and mixtures of S metabolites, containing also D7S and DDS. On the contrary, the mixture of gut metabolites, containing Eq and O-DMA, completely impairs D effects, resulting in estrogen mimetic action. Identical results have been obtained in both MCF-7 and T47D (not completely reported), suggesting that the effect is not dependent on the cellular context. These results highlight the need to use physiologically relevant metabolites when investigating the putative beneficial properties of polyphenols against cancer. 

## 4. Materials and Methods

### 4.1. Reagents

The Bradford protein assay and the chemiluminescence reagents for Western blot Clarity Western ECL Substrate were obtained from Bio-Rad Laboratories (Hercules, CA, USA). The anti-phospho-ERα (pERα Ser118) antibody and anti-phospho-AKT (pAKT Ser473) were purchased from Cell Signalling Technology Inc. (Beverly, MA, USA). The anti-α-tubulin and anti-NGB antibodies were purchased from Merck (Darmstadt, D). Specific antibodies against AKT, ERα (HC20), and poly [ADP-ribose] polymerase 1 (PARP-1) were obtained from Santa Cruz Biotechnology (Santa Cruz, CA, USA). The ERα inhibitor endoxifen was purchased from Tocris (Ballwin, MO, USA). All the other products were from Merck. Analytical and reagent grade products were used without further purification.

### 4.2. Cell Culture and Treatment

Human breast cancer cells MCF-7 and T47D (American Type Culture Collection, LGC Standards S.r.l., Milano, Italy) were grown in air containing 5% CO2 in either modified, phenol red free, Dulbecco’s Modified Eagle’s Medium (DMEM) medium. Ten percent (*v*/*v*) of charcoal-stripped fetal calf serum, L-glutamine (2 mM), gentamicin (0.1 mg/mL), and penicillin (100 U/mL) were added to the media before use. Cells were used at passage 4–8, as previously described [28]. Cell line authentication was periodically performed by the amplification of multiple short tandem repeat loci by BMR genomics S.r.l (Padova, Italy). Cells were treated for 24 h with either vehicle (DMSO/phosphate-buffered saline, 1:10; *v*/*v*) or E2 (1 or 10 nM) or D and/or its metabolites (0.1, 1.0, and 10 μM) or Pacl (100 nM) or the metabolites mixtures. Two different metabolite mixtures were used for cell treatment, Sulphate metabolites (S metab) and Gut metabolites (Gut metab). The mixture of S metab was constituted by D4S (1 µM), D7S (1 µM) and DDS (1 µM), while that of Gut metab was constituted by Equol (1 µM) and O-DMA (1 µM). When indicated, endoxifen (1 μM) was added 30 min before compound administration, or E2 (10 nM) was added 4 h before Pacl addition (0.1, 1, and 100 nM) for 24 h. The E2 concentrations were selected on the bases of dose-response experiments already performed [40]. 

### 4.3. Western Blot Assay

Briefly, after the treatments, cells were lysed and proteins were solubilized in the YY buffer (50 mM HEPES at pH 7.5, 10% glycerol, 150 mM NaCl, 1% Triton X-100, 1 mM EDTA, and 1 mM EGTA) containing 0.70% (*w*/*v*) sodium dodecyl sulfate (SDS). Total proteins were quantified using the Bradford protein assay. Solubilized proteins (20 μg) were resolved by 7% or 15% SDS-polyacrylamide gel electrophoresis at 100 V for 1 h at 24.0 °C and then transferred to nitrocellulose with the Trans-Blot Turbo Transfer System (Bio-Rad) for 7 min. The nitrocellulose was treated with 5% (*w*/*v*) bovine serum albumin in 138.0 mM NaCl, 25.0 mM Tris, pH 8.0, at 24.0 °C for 1 h. Nitrocellulose was probed overnight at 4.0 °C with either anti-NGB (final dilution, 1:1000), or anti-pERα (final dilution, 1:1000), or anti-pAKT (final dilution, 1:1000), or anti p-38 (final dilution 1:1000), or anti-PARP-1 (final dilution, 1:1000) antibodies. The nitrocellulose was stripped by Restore Western Blot Stripping Buffer (Pierce Chemical, Rockford, IL) for 10 min at room temperature and then probed with either anti-ERα (final dilution, 1:1000) or anti-AKT (final dilution 1:1000) or anti-p38 (final dilution, 1:1000) or anti-α-tubulin (final dilution, 1:30,000) antibodies to normalize the protein loaded. The antibody reactivity was detected with ECL chemiluminescence Western blotting detection reagent using a ChemiDoc XRS+ Imaging System (Bio-Rad Laboratories, Hercules, CA, USA). Densitometric analyses were performed by the ImageJ software for Microsoft Windows (National Institute of Health, Bethesda, Rockville, MD, USA).

### 4.4. Cellular DNA Content, Propidium Iodide (PI) Assay

Cells were grown up to 80% confluence in 96-well plate and treated with the selected compounds. The cells were fixed and permeabilized with cold ethanol (EtOH) 70% for 15 min at −20 °C. EtOH solution were removed and the cells were incubated with PI buffer for 30 min in the dark. The solution was removed, and the cells were rinsed with PBS solution. The fluorescence was revealed (excitation wavelength: 537 nm; emission wavelength: 621 nm) with TECAN Spark 20 M multimode microplate reader (Life Science, Milano, Italy).

### 4.5. Statistical Analysis

The statistical analysis was performed by Student’s t test to compare two sets of data by INSTAT software system for Windows. In all cases, *p* < 0.05 was considered significant.

## 5. Conclusions

Breast cancer, as well as lung, bronchial, and colorectal cancer, are estimated to be the three most commonly diagnosed types of malignancies. In particular, breast cancer alone accounts for 29% of all new cancers among women and the age of its onset is rapidly decreasing [41]. ERα activation status and signaling is considered one of the major drivers stimulating breast cancer proliferation, survival, and invasion [23,24,25]. The importance of ERα lies within its prognostic value, as it identifies patients that could benefit from the endocrine therapy [25]. Although the use of ERα antagonists has led to a considerable decline in breast cancer mortality, many patients become resistant to this therapy and developed metastatic tumors [24,41]. These observations have sparked the need for alternative approaches, increasing a sustained interest in soy isoflavones as a promising therapeutic option in breast cancer chemoprevention or as an adjuvant for chemotherapeutic agents. These claims lead patients with increased breast cancer risk to take into consideration supplementing their diet with soy or soy derivates, assuming that a high consumption might reduce the cancer risk [30,42]. Unfortunately, the increased economic interests in soy isoflavones has not yet been paralleled by the deciphering of the cellular and molecular mechanisms underlying their potential chemo-preventive role, which remains obscure. 

The data reported here strongly sustain the need to examine in deep the effect and the action mechanisms of soy isoflavone. In particular, the activity of gut microbiota should be investigated in patients before isoflavone consumption, due to its key role in the metabolism and bioavailability of isoflavones [43], microflora influencing factors also require consideration. As examples, a high-carbohydrate milieu causes more extensive biotransformation, greatly enhancing Eq formation, and a *Clostridium* sp. strain that converts D principally to O-DMA has been identified from gut microbiota. These bio-transformations could render isoflavones less active [44] and reduce the efficacy of the chemotherapeutic treatment. Moreover, concerns could arise also among healthy individuals who regularly consume soy products. Indeed, Equol and O-DMA producer individuals may be subjected to a long exposure to potent estrogenic compounds. Those who are not, on the other hand, will be mostly exposed to anti-estrogenic compounds.

## Figures and Tables

**Figure 1 cancers-12-00167-f001:**
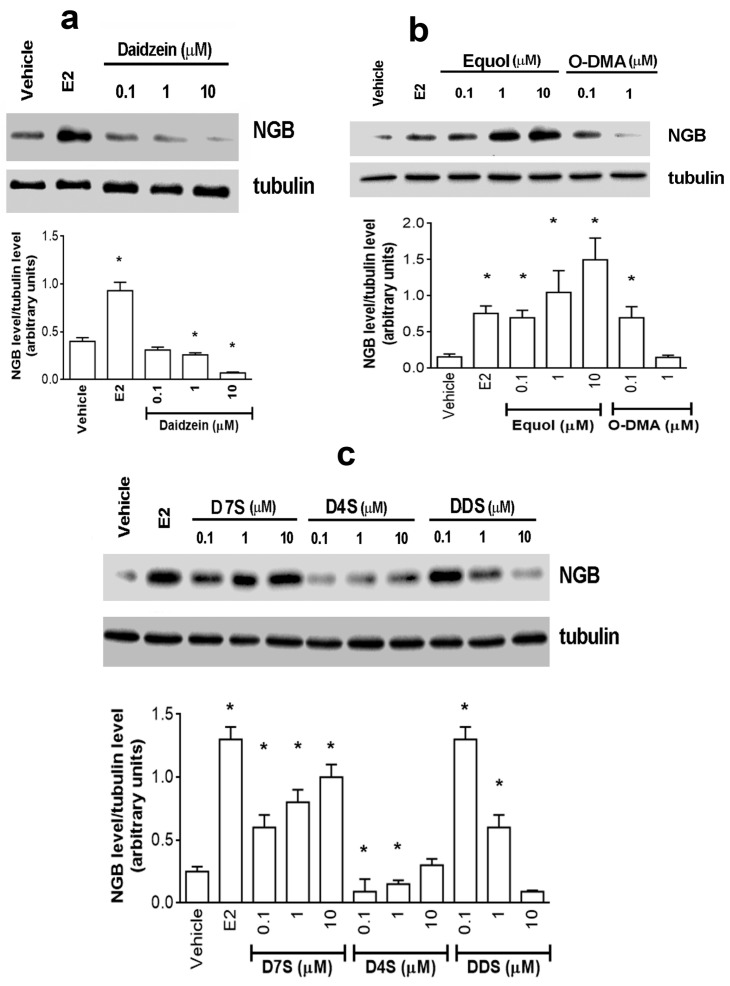
Effects of daidzein (**a**), equol and O-desmethylangolesin (**b**), daidzein-7-sulfate, daidzein-4′-sulfate and daidzein-7,4′-disulfate (**c**) on neuroglobin intracellular levels. (**a**–**c**) Western blot (top) and densitometric analyses (bottom) of NGB protein levels in MCF-7 cells treated for 24 h with the vehicle (DMSO), E2 (10 nM), D and its metabolites (0.1, 1.0, and 10 μM). The amount of proteins was normalized by comparison with tubulin levels. Data are the mean ± SD of four different experiments. *p* < 0.001 was determined with Student t test with respect to the vehicle (*) treated samples. E2: estradiol; NGB: neuroglobin; DMSO: dimethyl sulfoxide; D: daidzein; Eq: equol; O-DMA: O-desmethylangolesin; D7S: daidzein-7-sulfate; D4S: daidzein-4′-sulfate; DDS: daidzein-7,4′-disulfate.

**Figure 2 cancers-12-00167-f002:**
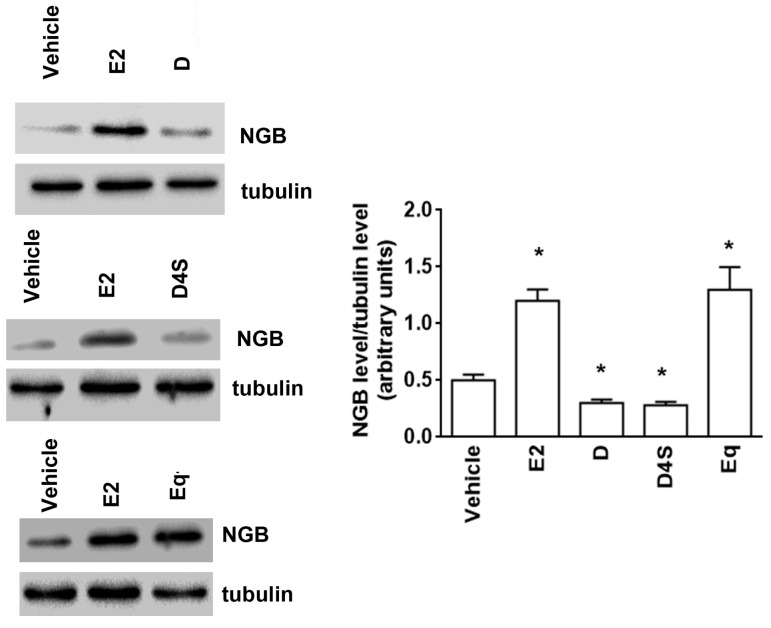
Effects of daidzein, daidzein-4′-sulfate and equol on neuroglobin intracellular levels in T47D cells. Western blot (left) and densitometric analyses (right) of NGB protein levels in T47D cells treated for 24 h with the vehicle (DMSO), E2 (10 nM), D (1 μM), D4S (1 μM), or Eq (1 μM). The amount of proteins was normalized by comparison with tubulin levels. Data are the mean ± SD of three different experiments. *p* < 0.001 was determined with Student t-test with respect to the vehicle (*) treated samples. DMSO: dimethyl sulfoxide; E2: estradiol; NGB: neuroglobin; D: daidzein; D4S: daidzein-4′-sulfate; Eq: equol.

**Figure 3 cancers-12-00167-f003:**
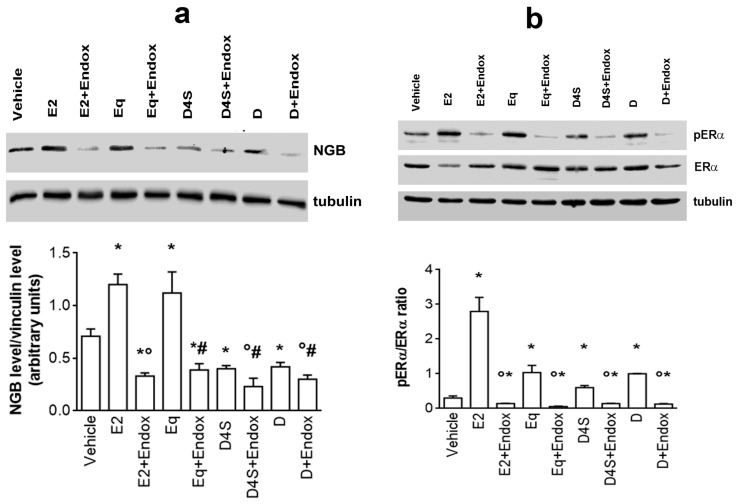
Daidzein, daidzein-4′-sulfate and equol effect on ERα activation status. (**a**) Western blot (top) and densitometric analyses (bottom) of NGB protein levels in MCF-7 cells treated for 24 h with either vehicle (DMSO) or E2 (10 nM) or D, D4S and Eq (1 μM) in presence or absence of the ERα inhibitor Endoxifen (1 μM; 30 min pretreatment). The amount of proteins was normalized by comparison with tubulin levels. Data are the mean ± SD of three different experiments. *p* < 0.001 was determined with Student’s t test with respect to the vehicle (*) or E2-treated (°) samples. (**b**) ERα activation by daidzein, daidzein-4′-sulfate and equol. The panel represents the ERαSer118 phosphorylation status calculated as the ratio pERα/ERα). Determined by Western blot analysis in MCF-7 cells exposed for 1h to either vehicle (DMSO) or E2 (10 nM) or D, D4S and Eq (1 μM) in presence or absence of ERα inhibitor Endoxifen (1 μM; 30 min pretreatment). The nitrocellulose was stripped and then probed with anti-ERα antibody. The pERα/ERα ratio was calculated with respect to tubulin obtained by densitometric analyses of three different experiments (mean ± SD). *p* < 0.001 was determined by Student t test with respect to vehicle (*), E2-treated (°) or Endox-untreated samples (#). DMSO: dimethyl sulfoxide; E2: estradiol; Endox: endoxifen; ERα: estrogen receptor α; NGB: neuroglobin; D: daidzein; D4S: daidzein-4′-sulfate; Eq: equol.

**Figure 4 cancers-12-00167-f004:**
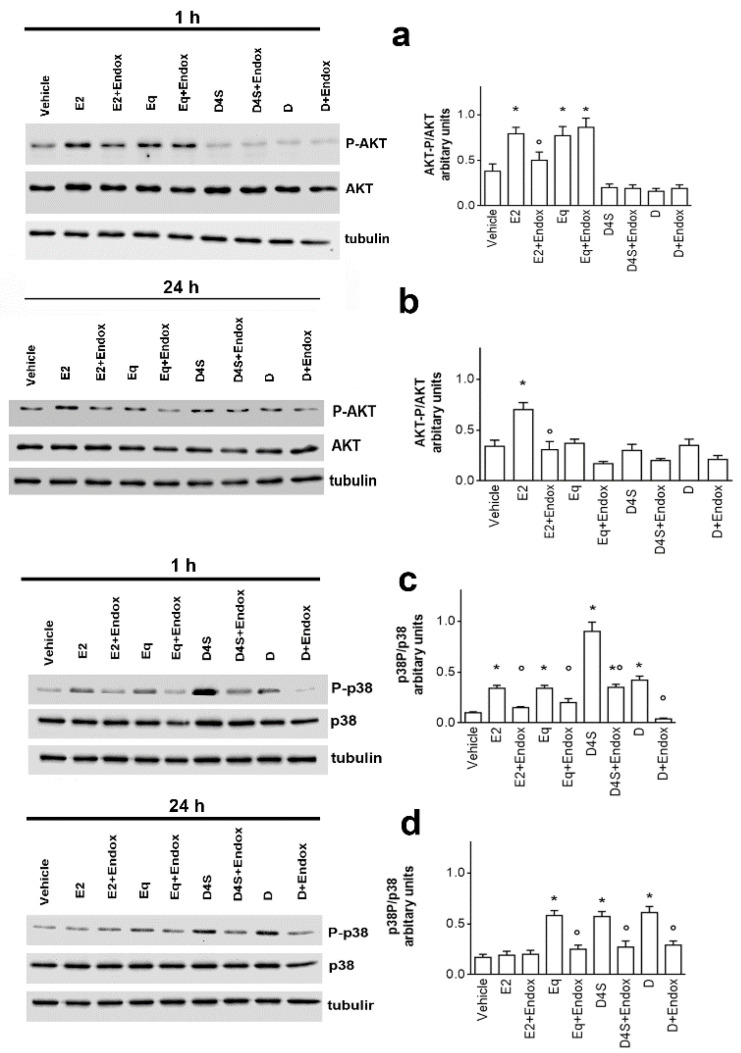
Daidzein, daidzein-4′-sulfate and equol action mechanism. The phosphorylation of the Ser473 residue of AKT (pAKT) (**a**,**b**) and Thr180/Tyr182 residues on P-38 (**c**,**d**) was determined by western blot analysis in MCF-7 cells exposed for 1 h (**a**,**c**) and 24 h (**b**,**d**) to either vehicle (DMSO:PBS 1:1) or E2 (10 nM) in presence or absence of D, D4S and Eq (1 μM). The nitrocellulose was stripped and then probed with anti-AKT or anti-p38 antibodies. In the panels, the PAKT/AKT (**a**,**b**) and Pp38/p38 (**c**,**d**) ratios are represented. These ratios are calculated with respect to tubulin obtained by densitometric analyses of three different experiments (mean ± SD). *p* < 0.001 was determined by Student t-test with respect to vehicle (*) or Endox-untreated (°) samples. AKT: protein kinase B; E2: estradiol; Endox: endoxifen; ERα: estrogen receptor α; NGB: neuroglobin; p38: p38 mitogen-activated protein kinase; D: daidzein; D4S: daidzein-4′-sulfate; Eq: equol.

**Figure 5 cancers-12-00167-f005:**
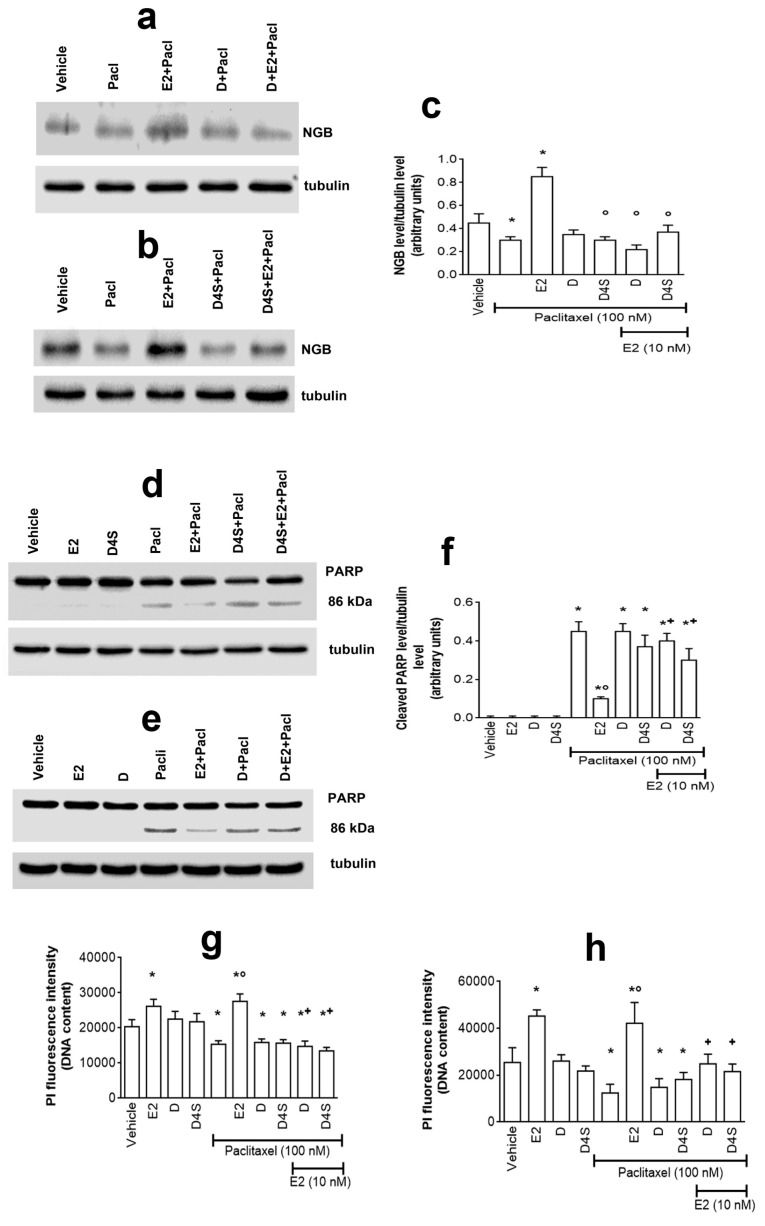
Physiological outcomes of D- and D4S-induced E2/ERα/NGB pathway avoidance. Western blot (left) and densitometric analyses (right) of NGB levels (**a**–**c**) and PARP-1 cleavage (**d**–**f**) in MCF-7 cells. Cells were treated with D (**a**,**e**) or D4S (**b**,**d**) (1 μM) in presence or absence of Pacl (100 nM) for 24 h, with either vehicle or E2; some Pacl treated cells were co-stimulated with either E2 or D or E2 together with D (**a**,**e**) or E2 or D4S or E2 together with D4S (**b**,**d**). (**c**,**f**) panels are densitometries that summarize the D- and D4S-induced modulation of NGB and PARP-1 cleavage respectively. The amount of protein was normalized by comparison with tubulin levels. Data are the mean ± SD of three different experiments. *p* < 0.001 was determined with Student t test with respect to the vehicle (*), Pacl-treated (°) samples or D- and D4S-treated samples, co-stimulated with Pacl but not with E2 (+). Effects of cellular DNA content obtained from PI assay on MCF-7 (**g**) or T47D (**h**) cells. The cells were treated for 24 h with either vehicle (DMSO) or E2 (10 nM) or Pacl (100 nM; 24 h); some samples were treated with D or D4S (1 µM) in presence or absence of Pacl and in presence of absence of E2. Data are mean ± SD of five different experiments. (*) *p* < 0.001 was calculated with Student t test versus vehicle or Pacl-treated (°) samples or D- and D4S-treated samples, co-stimulated with Pacl but not with E2 (+). E2: estradiol; NGB: neuroglobin; DMSO: dimethyl sulfoxide; D: daidzein; Eq: equol; D4S: daidzein-4′-sulfate; Pacl: paclitaxel.

**Figure 6 cancers-12-00167-f006:**
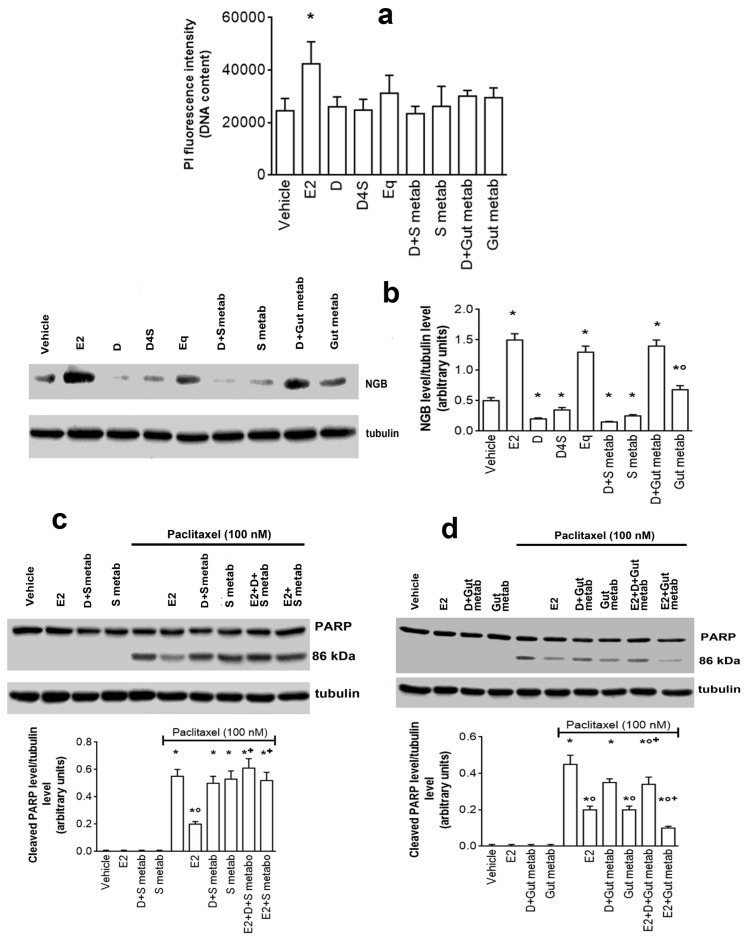
Physiological outcomes of D in mixture with its metabolites. (**a**) Analyses of cellular DNA content obtained from PI assay, Western blot (left) and densitometric analyses (right) of NGB levels (**b**) in MCF-7 cells. (**c**,**d**) Western blot (up) and densitometric analyses (bottom) of PARP-1 cleavage in MCF-7 cells. The MCF-7 cells were treated for 24 h with either vehicle (DMSO) or E2 (10 nM) or D, D4S or Eq (1 µM); some samples were treated with all the sulfate metabolites (D4S, D7S and DDS, 1 µM each) or all the gut metabolites (Eq and O-DMA, 1 µM each) in presence or absence of D (1 µM). Data are mean ± SD of three different experiments. (*) *p* < 0.001 was calculated with Student t test versus vehicle in (**a**). In **b**, data are the mean ± SD of five different experiments: *p* < 0.001 was determined with Student t test with respect to the vehicle (*) or to D-treated samples co-stimulated with metabolites (°). In (**c**,**d**), data are the mean ± SD of three different experiments. *p* < 0.001 was determined with Student *t* test with respect to the vehicle (*) or Pacl-treated (°) or E2 untreated samples co-stimulated with the metabolites (+). E2: estradiol; NGB: neuroglobin; DMSO: dimethyl sulfoxide; D: daidzein; Eq: equol; O-DMA: o-desmethylangolesin; D7S: daidzein-7-sulfate; D4S: daidzein-4′-sulfate; DDS: daidzein-7,4′-disulfate; Pacl: paclitaxel.

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
