# Peer review of "Divergent Effects of Daidzein and Its Metabolites on Estrogen-Induced Survival of Breast Cancer Cells"

_cancers, 2020, doi:10.3390/cancers12010167_

Round 1

Reviewer 1 Report

The present manuscript describes opposed results when treating breast cancer cell lines with the different metabolites derived from daidzein, a naturally occurring compound, structurally belonging to isoflavones, found in soybean.

Daidzein efficacy has been documented as an antiestrogen, in breast cancer. It seems it activates a pro-apoptotic cascade through the ER-beta receptor. But no clear results are known respect to its role on ER-beta receptor. As breast cancer has a positive ratio of ER-alpha respect to ER-beta receptors, the possible participation of daidzein interacting with ER-alpha should be elucidated. ER-alpha activity leads to PI3K/Akt activation and to final expression of the anti-apoptotic globin called neuroglobin, a molecule that will defend breast cancer cells against chemotherapy, like paclitaxel.

Daidzein is metabolized by the microbiota and by the liver, and their metabolites’ effects are unknown. Therefore, the aim of this study is trying to discover the role of these metabolites as estrogenic or antiestrogens, at the level of ER-alpha, and also trying to decipher whether any of those metabolites can (as resveratrol) interfere with ER-alpha estrogen induced neuroglobin up-regulation, thereby potentiating paclitaxel pro-apoptotic effects.

The study is well planned. Estrogen, daidzein, the different metabolites are given to cells. Also, an inhibitor of the estrogen receptor is used (endoxifen). The PI3K/Akt pathway and p38-MAPK pathway are explored. Also the expression of neuroglobin and PARP.

Finally, only one out of the five metabolites derived from daidzein, D4S, proved to have anti-estrogenic activity as daidzein. The article is important to show that not only daidzein, but also its metabolites, have to be taken into account when administering daidzein –as an anti-estrogenic drug- with the aim of targeting the ER-alpha receptor in breast cancer.

MAJOR QUESTIONS

Although the study seems to be presented as to be done in two breast cancer cell lines, most figures contain just informative results referred to MCF-7 cancer cells, while just Figure 2 presents results on T47D cells. If possible, new figures on T47D results should be presented. Otherwise, explain why only one cell line is represented in the study, while two cells lines were cited in Materials and Methods as to be used along the whole study.

Author Response

MAJOR QUESTIONS

Although the study seems to be presented as to be done in two breast cancer cell lines, most figures contain just informative results referred to MCF-7 cancer cells, while just Figure 2 presents results on T47D cells. If possible, new figures on T47D results should be presented. Otherwise, explain why only one cell line is represented in the study, while two cells lines were cited in Materials and Methods as to be used along the whole study.

Reply: We thank the referee for this question that allow us to clarify that all experiments have been performed in both MCF-7 and T47D cell lines. Identical effects of E2, daidzein and it metabolites have been obtained. Thus, to avoid to duplicate figures containing similar results, we choose to report just the T47D effects on  neuroglobin intracellular levels (fig. 2) and on the cellular DNA content obtained from PI assay (Fig. 5h). However, a general sentence explaining our choice has been added at the end of results (pg. 8 lines 207-208) and in the discussion section (pg. 11 lines 302-303). 

Moreover a careful check of misspelling  has been performed.

Reviewer 2 Report

The manuscript entitled “Divergent effects of daidzein and its metabolites on estrogen-induced survival of breast cancer cells” provides novel knowledge that may contribute to improve the treatment of breast cancer using daidzein and its derivertives. I think that data showed in this paper are approximately sound. I read the manuscript with interest, and I think that the results obtained in this paper may be had interest by many researchers who study the treatment of breast cancer. Especially, I think that the effects of metabolic transformation of isoflavones on estrogen-induced survival of breast cancer cells are very important to improve the treatment of breast cancer using them. Concerns raised are shown as below.

Concerns:

1) Regulatory mechanisms of NGB gene expression.

The authors showed only protein levels of NGB but not mRNA levels. Can the estrogen receptors activate transcription of the NGB gene as transcription factors? What roles of daidzein and its derivatives are in regulation of the gene expression of NGB gene? The authors should carry out RT-PCR in order to know the effects of estrogen, daidzein or its derivatives on transcription of NGB gene. Next, the authors should perform ChIP assay to study the roles of estrogen, daidzein or its derivatives in transcriptional regulation of NGB gene. I think that it is very important to understand the regulatory mechanisms of NGB gene expression by estrogen, daidzein or its derivatives.

2) Activation of caspases.  

The authors investigated the paclitaxel-mediated apoptosis in the breast cancer cells using PARP-1 cleavage as an apoptosis index. I think that the authors should identify the caspase responsible for the paclitaxel-mediated apoptosis.

3) About anti-cancer agents.

I think that the author should explain the reason why the authors used paclitaxel as an anti-cancer agent in this study. How about other anti-cancer agents against breast cancer?

4) Figure 4b

Data of E2 + Endox are not shown in Figure 4b.

5) Results 2.2.

The authors described "Endoxifen pretreatment prevents the D and D4S effects as well as that of E2, although 1h after D4S stimulation the ER inhibitor does not completely impede p38 activation, suggesting that an ERα-independent mechanism is at the root of the very high p38 phosphorylation induced by this sulphate metabolite". Where are the data shown?

Author Response

We thank the reviewer for her/his helpful Comments and Suggestions

Concerns:

1) Regulatory mechanisms of NGB gene expression. The authors showed only protein levels of NGB but not mRNA levels. Can the estrogen receptors activate transcription of the NGB gene as transcription factors? What roles of daidzein and its derivatives are in regulation of the gene expression of NGB gene? The authors should carry out RT-PCR in order to know the effects of estrogen, daidzein or its derivatives on transcription of NGB gene. Next, the authors should perform ChIP assay to study the roles of estrogen, daidzein or its derivatives in transcriptional regulation of NGB gene. I think that it is very important to understand the regulatory mechanisms of NGB gene expression by estrogen, daidzein or its derivatives.

REPLY: the mechanisms at the root of E2-induced NGB accumulation in breast cancer cells has been already defined in our previous papers cited in the manuscript (Fiocchetti et al., 2018), whereas the transcriptional regulation of NGB gene has been described by other Authors (Guglielmotto M, et al.,  Front Cell Neurosci. 2016 Cutrupi S, et al., IUBMB Life. 2014 Jan;66(1):46-51). Briefly, E2 stimulation rapidly induced ERα-mediated AKT phosphorylation that, in turn, affects NGB degradation (block of proteasomal and lysosomal degradation) assuring the rapid NGB accumulation into the cell. In parallel, AKT activation (via CREBP activation) is necessary to assure NGB transcription that allows further accumulation of protein. Thus, E2 induces the indirect transcription of NGB; indeed NGB promoter does not contain any estrogen responsive element. On the bases of these data, we focused our study on AKT because of just AKT activating compounds (e.g., equol) could assure NGB accumulation.

2) Activation of caspases. The authors investigated the paclitaxel-mediated apoptosis in the breast cancer cells using PARP-1 cleavage as an apoptosis index. I think that the authors should identify the caspase responsible for the paclitaxel-mediated apoptosis.

REPLY: It is well known that paclitaxel could activate both caspase 3/7 and caspase 8 in breast cancer (Quispe-Soto and Calaf Int J Oncol. 2016 Dec;49(6):2569-2577; Calaf et al., Oncol Rep. 2018 Oct;40(4):2381-2388). Moreover, NGB is involved in preventing mitochondrial-dependent (intrinsic) apoptotic pathway that involves caspase 3/7 activation. MCF-7 does not express caspase-3, thus, only caspase 7 can be involved. For this reason, the cell death has been evaluated with two different approaches: the Propidium assay (number of viable cells) and the cleavage of PARP-1 (direct apoptotic marker).

3) About anti-cancer agents. I think that the author should explain the reason why the authors used paclitaxel as an anti-cancer agent in this study. How about other anti-cancer agents against breast cancer?

REPLY: Paclitaxel, a first-line therapeutic clinical agent used to treat breast cancer, exerts its antitumor activity by promoting the polymerization of tubulin and stabilizing the resulting microtubules, causing cell cycle arrest that leads to apoptosis of cancer cells. A wide literature is available on the mechanisms at the root of paclitaxel effect, including estrogen-induced paclitaxel resistance,  that render this compound a prototype of chemotherapeutic agent for breast cancer. A sentence has been added in the discussion section (page 11 lines 282-283).

4) Figure 4b. Data of E2 + Endox are not shown in Figure 4b.

REPLY: The figure 4b has been replaced with one reporting all treatments

5) Results 2.2. The authors described "Endoxifen pretreatment prevents the D and D4S effects as well as that of E2, although 1h after D4S stimulation the ER inhibitor does not completely impede p38 activation, suggesting that an ERα-independent mechanism is at the root of the very high p38 phosphorylation induced by this sulphate metabolite". Where are the data shown?

REPLY: the data are reported in fig. 4c that now has been cited in the text.

Reviewer 3 Report

cancers-667859

Manuscript ID: cancers-667859

Type of manuscript: Article
Title: Divergent effects of daidzein and its metabolites on  estrogen-

induced survival of breast cancer cells

To Authors:

This study investigates the different effects of daidzein and its metabolities, derived from liver or gut bacteria, on breast cancer cells. Authors observed that contradictory effects of these drugs on survival of breast cancer cells. This is an interesting points in studying phytomedicine drugs in cancer cells treatment. Some of minor questions are list below.

Minor:

Page 2, line 81: 1 mM concentration. In the legend of figure 1, 0.1 ~ 10 mM has been listed?

Page 7, Line 187~192: Why do authors choose the concentrations for these metabolities? From reference?

Page 9, line 243~249: Is there the values for each metabolites in plasma?

         Line 255: ant? → and?

Author Response

We thank the reviewer for her/his helpful Comments and Suggestions

Minor:

Page 2, line 81: 1 mM concentration. In the legend of figure 1, 0.1 ~ 10 mM has been listed?

REPLY: the concentration 1µM was selected for successive experiments. The text was amended accordingly.

Page 7, Line 187~192: Why do authors choose the concentrations for these metabolities? From reference?

REPLY: The metabolites concentrations were selected from the results reported in figure 1, while the concentration for E2 and paclitaxel were selected from literature.

Page 9, line 243~249: Is there the values for each metabolites in plasma?

REPLY: This is an interesting question. While the plasma concentration of isoflavone aglicones has been determined (usually by HLPC followed by mass), as far we known the plasma level of its metabolites is more difficult to determine depending by the differences in the metabolism of every individuals. Thus, the reported references give a possible percentage with respect to the precursors not a precise value. 

Line 255: ant? → and?

REPLY: The sentence has been amended

Reviewer 4 Report

The manuscript by Montalesi et al 2019 provides an insight into the effects of diadzein and some of its metabolites on estrogen-induced survival of MCF-7 and T47D breast cancer cell lines. The authors first consider the effects of these agents upon neuroglobin levels evaluated by Western blotting Figure 1).  The blotting work is clear with data representative of 4 independent experiments.  However, the authors chose two concentrations (1 and 10 µM) and firstly need to relate these concentrations to circulatory levels that might be expected in vivo, but moreover, it would be advantageous to demonstrate that the effects are concentration dependent, in which case 3 more concentrations would be required.  Lastly, as there are multiple concentrations, I was surprised that the authors did not perform an ANOVA, rather than singularly comparing each concentration with the vehicle control.

For Figures 3 and 4– the authors considered the relative levels of phosphorylated proteins vs total proteins. This is a useful insight to the signalling of the agents. Just as a technical issue, the authors assessed phospho forms and then stripped the nitrocellulose membrane and then reprobed with the total protein antibody.  Although stripping and reprobing can work ok and allows a direct comparison of the same phospho form with total protein, it also assumes an even strip across the surface of the blot, and this may not be the case.  With this regard, studies sometimes also consider alternative (or additional) means to normalise protein via total protein staining in lanes, and certainly this could have been performed post-stripping of the membrane. The authors may also locate antibody pairs that could be used simultaneously to evaluate phospho and non-phosphorylated protein forms.

More details that relate the concentrations of the metabolites to those that might be expected to circulate in vivo needs to be included.   Furthermore, as detailed above, it would be more useful to consider the cellular effects if a more complete concentration response is included – currently only a single concentration is considered.  A further concern with section 2.4 is that the metabolites could be cytostatic rather than cytotoxic, and just limit the proliferation of the cells.  The authors could run tandem BrdU assays, or even MTT/LDH assays to consider the true ‘cytotoxic’ nature of these chemicals.

One final comment about methodology is that the authors should detail which protein(s) were used for their protein standard curve. There are some reservations with using Bradford reagent for protein concentration determinations (section 4.3) with high detergent concentrations, whereas the commercially available modified Lowry methods have some detergent compatibility and may therefore prove to be more reliable for the lysis conditions employed by the authors.

Lastly, there are a few wording changes that I would recommend. The abstract details “among the 5 metabolites”, but this is not informative to the reader as little detail about the metabolites are included in the abstract.  The term physiological (2.3, 2.4, Figure 5 etc) is inappropriate since ultimately this study is solely in vitro, there are no physiological or indeed pathological in vivo experiments performed.  I would therefore suggest this terminology is revised or omitted from the text.  Within the discussion, the authors refer to dose-dependent effects and a dose-response curve obtained.  This is not technically correct, the term dosing should only be used for animals (and human) studies so only truly relates to in vivo work.  The authors should change this wording to concentration dependent and concentration curve etc.  However, with that said, to perform a true concentration curve at least 5 different concentrations should be performed.

Author Response

We thank the reviewer for her/his helpful Comments and Suggestions

The manuscript by Montalesi et al 2019 provides an insight into the effects of diadzein and some of its metabolites on estrogen-induced survival of MCF-7 and T47D breast cancer cell lines. The authors first consider the effects of these agents upon neuroglobin levels evaluated by Western blotting Figure 1).  The blotting work is clear with data representative of 4 independent experiments.  However, the authors chose two concentrations (1 and 10 µM) and firstly need to relate these concentrations to circulatory levels that might be expected in vivo, but moreover, it would be advantageous to demonstrate that the effects are concentration dependent, in which case 3 more concentrations would be required.  Lastly, as there are multiple concentrations, I was surprised that the authors did not perform an ANOVA, rather than singularly comparing each concentration with the vehicle control.

REPLY: Apart from O-DMA (0.1 and 1µM), Figure 1 reports 3 different concentrations for all compounds ranging from 0.1 to 10 µM (i.e., 0.1, 1.0, and 10 µM) being 1.0 µM the plasma concentration found after a meal reach in soy (1-5 µM). The aim of this experiment is to find the more effective concentration of each compound able to give an effect, for this reason we prefer Student’s t test comparing each concentration with the control.

For Figures 3 and 4– the authors considered the relative levels of phosphorylated proteins vs total proteins. This is a useful insight to the signalling of the agents. Just as a technical issue, the authors assessed phospho forms and then stripped the nitrocellulose membrane and then reprobed with the total protein antibody.  Although stripping and reprobing can work ok and allows a direct comparison of the same phospho form with total protein, it also assumes an even strip across the surface of the blot, and this may not be the case.  With this regard, studies sometimes also consider alternative (or additional) means to normalise protein via total protein staining in lanes, and certainly this could have been performed post-stripping of the membrane. The authors may also locate antibody pairs that could be used simultaneously to evaluate phospho and non-phosphorylated protein forms.

REPLY: We are sure that there are different methodological approaches to obtain results, in particular, all Authors completely agree with the sentence stated by the referee ‘the relative levels of phosphorylated proteins vs total proteins … is a useful insight to study the signalling’. Moreover, all Authors agree with the successive referee’s sentence ‘stripping and reprobing can work ok and allows a direct comparison of the same phospho form with total protein’. However, it is interesting to compare different approaches, thus we thank the referee for its comments.

More details that relate the concentrations of the metabolites to those that might be expected to circulate in vivo needs to be included.  Furthermore, as detailed above, it would be more useful to consider the cellular effects if a more complete concentration response is included – currently only a single concentration is considered. 

REPLY: As reported in the reply to referee 3, while the plasma concentration of isoflavone aglicones has been determined (usually by HLPC followed by mass), as far we known the plasma level of its metabolites is more difficult to determine depending by the differences in the metabolism of every individuals. More data are available on the urinary excretion of isoflavones and some, but not all, of its metabolites (principally equol). Thus, it is difficult to detail on the possible plasma concentrations of daidzein metabolites. This aspect requires more experiments that are expensive and difficult to perform for compounds that do not have yet any clear effect on human health. Our hope is that reporting the cellular effects of these compounds could prompt researchers to fill this gap of knowledge.

A further concern with section 2.4 is that the metabolites could be cytostatic rather than cytotoxic, and just limit the proliferation of the cells.  The authors could run tandem BrdU assays, or even MTT/LDH assays to consider the true ‘cytotoxic’ nature of these chemicals.

REPLY: Aim of this paper was to determine if daidzein metabolites maintain the same antiestrogenic effect of their precursor. The concern raised by referee, although interesting, drive the paper far from its original aim addressing the effects of the daiadzein metabolites that may be not related to their anti-estrogenicity.  

One final comment about methodology is that the authors should detail which protein(s) were used for their protein standard curve. There are some reservations with using Bradford reagent for protein concentration determinations (section 4.3) with high detergent concentrations, whereas the commercially available modified Lowry methods have some detergent compatibility and may therefore prove to be more reliable for the lysis conditions employed by the authors.

REPLY: We used Albumin fraction V for our standard curve. We used original Lowry methods for years, than after the opportune check we decide to use Bradford (less expensive and less time consuming). We agree with the referee opinion, but protein determination is used just to be sure that the same quantity of each protein is loaded on electrophoresis gel. If a SDS perturbation occurs this will be the same for all cell lysates that contain the same quantity of detergent.

Lastly, there are a few wording changes that I would recommend. The abstract details “among the 5 metabolites”, but this is not informative to the reader as little detail about the metabolites are included in the abstract. 

REPLY: the sentence has been modified

The term physiological (2.3, 2.4, Figure 5 etc) is inappropriate since ultimately this study is solely in vitro, there are no physiological or indeed pathological in vivo experiments performed.  I would therefore suggest this terminology is revised or omitted from the text. 

REPLY: One of the main topic of physiology is Cell Physiology, which object of study is the cell responses to external signals that could challenge their homeostasis. Our lab is focused on this issue since many years studying the cell response to hormones and the possible perturbation of these responses by diet-derived compounds or contaminants. Although in vivo experiments allow to obtain more holistic vision of organismic responses, experiments focused on cell responses are considered physiological.    

Within the discussion, the authors refer to dose-dependent effects and a dose-response curve obtained.  This is not technically correct, the term dosing should only be used for animals (and human) studies so only truly relates to in vivo work.  The authors should change this wording to concentration dependent and concentration curve etc.  However, with that said, to perform a true concentration curve at least 5 different concentrations should be performed.

REPLY: In our opinion, the concept of dose-response is not applicable only to the animal or human in vivo experiments. Indeed, the dose–response relationship describes the magnitude of the response of an organism or a cell or a molecule, as a function of concentration (or doses) to a stimulus after a certain exposure time. The dose–response curves describes the dose–response relationships. This concept is used in endocrinology, physiology, pharmacology, and toxicology. However, we change the term dose-response in the discussion with concentrations as in the other part of the text.

Round 2

Reviewer 2 Report

My questions and comments on the manuscript are now responded.

Reviewer 4 Report

The authors have provided a satisfactory response to the revisions requested.  I would suggest accept for publication.